# PAC-Bayes Generalisation Bounds for Heavy-Tailed Losses through Supermartingales

**Maxime Haddouche** *maxime.haddouche@inria.fr*
*Inria, University College London and Université de Lille*

**Benjamin Guedj** *benjamin.guedj@inria.fr*
*Inria and University College London*

**Reviewed on OpenReview:** *https://openreview.net/forum?id=qxrwt6F3sf*

## Abstract

While PAC-Bayes is now an established learning framework for light-tailed losses (*e.g.*, sub-gaussian or subexponential), its extension to the case of heavy-tailed losses remains largely uncharted and has attracted a growing interest in recent years. We contribute PAC-Bayes generalisation bounds for heavy-tailed losses under the sole assumption of bounded variance of the loss function. Under that assumption, we extend previous results from Kuzborskij and Szepesvári (2019). Our key technical contribution is exploiting an extention of Markov's inequality for supermartingales. Our proof technique unifies and extends different PAC-Bayesian frameworks by providing bounds for unbounded martingales as well as bounds for batch and online learning with heavy-tailed losses.

## 1 Introduction

PAC-Bayes learning is a branch of statistical learning theory aiming to produce (tight) generalisation guarantees for learning algorithms, and as a byproduct leads to designing new efficient learning procedures by minimising such guarantees. Generalisation bounds are helpful for understanding how a learning algorithm may perform on future similar batches of data. Since its emergence in the late 1990s, PAC-Bayes theory (see the seminal works of Shawe-Taylor and Williamson, 1997; McAllester, 1998; 1999; 2003; Catoni, 2003; 2007) has revealed powerful enough to explain the generalisation abilities of learning algorithms which output distributions over the predictors space (a particular case being when the output is a Dirac mass on a single predictor) from which our predictors of interest are designed. We refer to the recent surveys from Guedj (2019); Alquier (2021) for an overview on PAC-Bayes.

At first, PAC-Bayes theory was mainly focused on classification tasks (see Seeger, 2002; Langford and Schapire, 2005; Catoni, 2007) but has quickly been generalised to any bounded loss function in regression (see *e.g.*, Maurer, 2004; Germain et al., 2009; 2016). PAC-Bayes learning covers a broad scope of domains and tools, from information theory (Kakade et al., 2008; Wu and Seldin, 2022) to statistical learning (Catoni, 2003; 2007), convex optimisation (Thiemann et al., 2017), Bernstein-type concentration inequalities (Tolstikhin and Seldin, 2013; Mhammedi et al., 2019), margins (Biggs and Guedj, 2022a; Biggs et al., 2022) and martingales (Seldin et al., 2011; 2012a;b), to name but a few.

From a practical perspective, the above led to generalisations guarantees for PAC-Bayes-inspired neural networks (NN): a promising line of work initiated by Dziugaite and Roy (2017) and pursued further by Letarte et al. (2019); Rivasplata et al. (2019); Pérez-Ortiz et al. (2021); Biggs and Guedj (2021); Perez-Ortiz et al. (2021a;b); Biggs and Guedj (2022b), among others, established NN architectures enjoying tight generalisation guarantees, turning PAC-Bayes into a powerful tool to handle complex neural structures (*e.g.*, Chérief-Abdellatif et al., 2022).

These encouraging breakthroughs gave rise to several initiatives to extend PAC-Bayes beyond the bounded losses assumption which is limiting in practice. Indeed, the goal is to make PAC-Bayes able to provide efficiency guarantees to any learning algorithm attached to a loss function. For instance, consider a NN trained to solve regression problems without constraints on the training domain. Several works already proposed routes to overcome the boundedness constraint: Catoni (2004, Chapter 5) already proposed PAC-Bayes bounds for the classification tasks and regressions ones with quadratic loss under a subexponential assumption. This technique has later been exploited in Alquier and Biau (2013) for the single-index model, and by Guedj and Alquier (2013) for nonparametric sparse additive regression, both under the assumption that the noise is subexponential. However all these works are dealing with light-tailed losses. Alquier and Guedj (2018); Holland (2019); Kuzborskij and Szepesvári (2019); Haddouche et al. (2021) proposed extensions beyond light-tailed losses. Our work stands in the continuation of this spirit while developing and exploiting a novel technical toolbox. To better highlight the novelty of our approach, we first present the two classical building blocks of PAC-Bayes.

## 1.1 Understanding PAC-Bayes: a celebrated route of proof

### 1.1.1 Two essential building blocks for a preliminary bound

In PAC-Bayes, we typically assume access to a non-negative loss function $\ell(h, z)$ taking as argument a predictor $h \in \mathcal{H}$ and data $z \in \mathcal{Z}$ (think of $z$ as a pair input-output $(x, y)$ for supervised learning problems, or as a single datum $x$ in unsupervised learning). We also assume access to a $m$-sized sample $S = (z_1, ..., z_m) \in \mathcal{Z}^m$ of data on which we will learn a meaningful posterior distribution $Q$ on $\mathcal{H}$, from a prior $P$ (or reference measure – see *e.g.*, Guedj, 2019 for a discussion on the terminology of probability distributions in PAC-Bayes).

To do so, PAC-Bayesian proofs are built upon two cornerstones. The first one is the change of measure inequality (Csiszár, 1975; Donsker and Varadhan, 1975; Dupuis and Ellis, 2011 – see also Banerjee, 2006; Guedj, 2019 for a proof).

**Lemma 1.1** (Change of measure inequality). *For any measurable function* $\psi : \mathcal{H} \to \mathbb{R}$ *and any distributions* $Q, P$ *on* $\mathcal{H}$:

$$\mathbb{E}_{h \sim Q}[\psi(h)] \leq \mathrm{KL}(Q, P) + \log\left(\mathbb{E}_{h \sim P}[\exp(\psi(h))]\right),$$

*with* $\mathrm{KL}$ *denoting the Kullback-Leibler divergence.*

The change of measure inequality is then applied to a certain function $f_m : \mathcal{Z}^m \times \mathcal{H} \to \mathbb{R}$ of the data and a candidate predictor: for all posteriors $Q$,

$$\mathbb{E}_{h \sim Q}[f_m(S, h)] \leq \mathrm{KL}(Q, P) + \log\left(\mathbb{E}_{h \sim P}[\exp(f_m(S, h))]\right). \tag{1}$$

To deal with the random variable $X(S) := \mathbb{E}_{h \sim P}[\exp(f_m(S, h))]$, our second building block is Markov's inequality $\left(\mathbb{P}(X > a) \leq \frac{\mathbb{E}[X]}{a}\right)$ which we apply for a fixed $\delta \in ]0, 1[$ on $X(S)$ with $a = \mathbb{E}_S[X(S)]/\delta$. Taking the complementary event gives that for any $m$, with probability at least $1 - \delta$ over the sample $S$, $X(S) \leq \mathbb{E}_S[X(S)]/\delta$, thus:

$$\mathbb{E}_{h \sim Q}[f_m(S, h)] \leq \mathrm{KL}(Q, P) + \log(1/\delta) + \log\left(\mathbb{E}_{h \sim P}\mathbb{E}_S[\exp(f_m(S, h))]\right). \tag{2}$$

### 1.1.2 From preliminary to complete bounds

From the preliminary result of Eq. (2), there exists several ways to obtain PAC-Bayesian generalisation bounds, all being tied to specific choices of $f$ and the assumptions on the dataset $S$. However they all rely on the control of an exponential moment implied by Markov's inequality: this is a strong constraint which has been at the heart of the classical assumption appearing in PAC-Bayes learning. For instance, the celebrated result of McAllester (1999), tightened by Maurer (2004), exploits in particular, a data-free prior, an iid assumption on $S$ and a light-tailed loss function. Most of the existing results stand with those assumptions (see *e.g.*, Catoni, 2007; Germain et al., 2009; Guedj and Alquier, 2013; Tolstikhin and Seldin,

2013; Guedj and Robbiano, 2018; Mhammedi et al., 2019; Wu and Seldin, 2022). Indeed, in many of these works, a boundedness assumption on the loss is used but in many cases, it can be relaxed to subgaussiannity without loss of generality. Catoni (2004) extended PAC-Bayes learning to the subexponential case. Many works tried to mitigate at least one of the following three assumptions.

- **Data-free priors.** With an alternative set of techniques, Catoni (2007) obtained bounds with localised (*i.e.*, data-dependent) priors. More recently, Lever et al. (2010); Parrado-Hernández et al. (2012); Lever et al. (2013); Oneto et al. (2016); Dziugaite and Roy (2017); Mhammedi et al. (2019) also obtained PAC-Bayes bound with data-dependent priors.

- **The iid assumption on *S*.** The work of Fard and Pineau (2010) established links between reinforcement learning and PAC-Bayes theory. This naturally led to the study of PAC-Bayesian bound for martingales instead of iid data (Seldin et al., 2011; 2012a;b).

- **Light-tailed loss.** PAC-Bayes bounds for heavy-tailed losses (*i.e.*, without subgaussian or subexponential assumptions) have been studied. Audibert and Catoni (2011) provide PAC-Bayes bounds for least square estimators with heavy-tailed random variables. Their results was suboptimal with respect to the intrinsic dimension and was followed by further works from Catoni and Giulini (2017) and Catoni (2018). More recently, this question has been adressed in the works of Alquier and Guedj (2018); Holland (2019); Kuzborskij and Szepesvári (2019); Haddouche et al. (2021), extending PAC-Bayes to heavy-tailed losses under additional technical assumptions.

Several questions then legitimately arise.

**Can we avoid these three assumptions simultaneously?** The answer is yes: for instance the work of Rivasplata et al. (2020) proposed a preliminary PAC-Bayes bound holding with none of the three assumptions listed above. Building on their theorem, Haddouche and Guedj (2022) only exploited a bounded loss assumption to derive a PAC-Bayesian framework for online learning, requiring no assumption on data and allowing data (history in their context)-dependent priors.

**Can we obtain PAC-Bayes bounds without the change of measure inequality?** Yes, for instance Alquier and Guedj (2018) proposed PAC-Bayes bounds involving $f$-divergences and exploiting Holder's inequality instead of Lemma 1.1. More recently, Picard-Weibel and Guedj (2022) developed a broader discussion about generalising the change of measure inequality for a wide range of $f$-divergences. We note also that Germain et al. (2009) proposed a version of the classical route of proof stated above avoiding the use of the change of measure inequality. This comes at the cost of additional technical assumptions (see Haddouche et al., 2021, Theorem 1 for a statement of the theorem in a proper measure-theoretic framework).

**Can we avoid Markov's inequality?** We mentioned above that several works avoided the change of measure inequality to obtain PAC-Bayesian bounds, but can we do the same with Markov's inequality? This point is interesting as avoiding Markov allow us to avoid assumptions such as sub-gaussiannity to provide PAC-Bayes bound. The answer is yes but this is a rare breed. To the best of our knowledge, only two papers are explicitly not using Markov's inequality: Kakade et al. (2008) obtained a PAC-Bayes bound using results on Rademacher complexity based on the McDiarmid concentration inequality, and Kuzborskij and Szepesvári (2019) exploited a concentration inequality from De la Peña et al. (2009), up to a technical assumptions to obtain results for unbounded losses. Both of this works did not required a bound on an exponential moment to hold.

## 1.2 Originality of our approach

Avoiding Markov's inequality appears challenging in PAC-Bayes but leads to fruitful results as those in Kuzborskij and Szepesvári (2019).

In this work, we exploit a generalisation of Markov's inequality for supermartingales: Ville's inequality (as noticed by Doob 1939). This result has, to our knowledge, never been used in PAC-Bayes before.

**Lemma 1.2** (Ville's maximal inequality for supermartingales)**.** *Let $(\mathcal{F}_t)$ be a filtration and $(Z_t)$ a non-negative super-martingale satisfying $Z_0 = 1$ a.s. If $Z_t$ is adapted to $\mathcal{F}_t$ and $\mathbb{E}[Z_t \mid \mathcal{F}_{t-1}] \leq Z_{t-1}$ a.s., $t \geq 1$, then, for any $0 < \delta < 1$, it holds*

$$\mathbb{P}\left(\exists T \geq 1 : Z_T > \delta^{-1}\right) \leq \delta.$$

*Proof.* We apply the optional stopping theorem (Durrett, 2019, Thm 4.8.4) with Markov's inequality defining the stopping time $i = \inf\{t > 1 : Z_t > \delta^{-1}\}$ so that

$$\mathbb{P}\left(\exists t \geq 1 : Z_t > \delta^{-1}\right) = \mathbb{P}\left(Z_i > \delta^{-1}\right) \leq \mathbb{E}[Z_i]\,\delta \leq \mathbb{E}[Z_0]\,\delta \leq \delta.$$

$\square$

A major interest of Ville's result is that it holds for a countable sequence of random variables simultaneously. This point is new in PAC-Bayes as it will allow us to obtain bounds holding for a countable (not necessarily finite) dataset $S = (z_i)_{i \geq 1}$.

**On which supermartingale do we apply Ville's bound ?** To fully exploit Lemma 1.2, we now take a countable dataset $S = (z_i)_{i \geq 1} \in \mathcal{Z}^m$. Recall that, because we use the change of measure inequality, we have to deal with the following exponential random variable appearing in Eq. (1) for any $m \geq 1$:

$$Z_m := \mathbb{E}_{h \sim P}[\exp(f_m(S, h))].$$

Our goal is to choose a sequence of functions $f_m$ such that $(Z_m)_m$ is a supermartingale. A way to do so comes from Bercu and Touati (2008).

**Lemma 1.3.** *Let $(M_m)$ be a locally square-integrable martingale with respect to the filtration $(\mathcal{F}_m)$. For all $\eta \in \mathbb{R}$ and $m \geq 0$, one has:*

$$\mathbb{E}\left[\exp\left(\eta \Delta M_m - \frac{\eta^2}{2}\left(\Delta[M]_m + \Delta\langle M\rangle_m\right)\right) \mid \mathcal{F}_{m-1}\right] \leq 1,$$

*where $\Delta M_m = M_m - M_{m-1}, \Delta[M]_m = \Delta M_m^2$ and $\Delta\langle M\rangle_m = \mathbb{E}\left[\Delta M_m^2 \mid \mathcal{F}_{m-1}\right]$.*

*We define $V_m(\eta) = \exp\left(\eta M_m - \frac{\eta^2}{2}\left([M]_m + \langle M\rangle_m\right)\right)$. Then, for all $\eta \in \mathbb{R}, (V_m(\eta))$ is a positive super-martingale with $\mathbb{E}[V_m(\eta)] \leq 1$ where $[M]_m(h) := \sum_{i=1}^m \Delta[M]_m, \langle M\rangle_m(h) := \sum_{i=1}^m \Delta\langle M\rangle_m$.*

In the sequel, this lemma will be helpful to design a supermartingale (*i.e.*, to choose a relevant $f_m$ for any $m$) without further assumption.

## 1.3 Contributions and outline

By avoiding Markov, a key message of (Kuzborskij and Szepesvári, 2019) is that, for learning problems with independent data, PAC-Bayes learning only requires the control of order 2 moment on losses to be used with convergence guarantees. This is strictly less restrictive than the classical subgaussian/subgamma assumptions appearing in the major part of the literature.

We successfully prove this fact remains even for non-independent data: we only need to control order 2 (conditional) moments to perform PAC-Bayes learning. Furthermore, our proof technique is general enough to encompass two different PAC-Bayesian framework: PAC-Bayes for martingales (Seldin et al., 2011; 2012a;b) and Online PAC-Bayes learning (Haddouche and Guedj, 2022). Thus, our main contributions are twofold.

- We provide a novel PAC-Bayesian bound holding for data-free priors and unbounded martingales. From this, we recover in PAC-Bayes bounds for unbounded losses and iid data as a significant particular case.

- We extend the Online PAC-Bayes framework of Haddouche and Guedj (2022) to the case of unbounded losses.

More precisely, Sec. 2.1 contains our novel PAC-Bayes bound for unbounded martingales and Sec. 2.3 contains an immediate corollary for learning theory with iid data. Our second contribution lies in Sec. 3 and extend Online PAC-Bayes theory to the case of unbounded losses. We eventually apply our main result for martingales in Sec. 4 to the setting of multi-armed bandit. Doing so, we provably extend a result of Seldin et al. (2012a) to the case of unbounded rewards.

Appendix A gathers more details on PAC-Bayes, we draw in Appendix B a detailed comparison between our new results and a few classical ones. We show that adapting our bounds to the assumptions made in those papers allows to recover similar or improved bounds. We defer to Appendix C the proofs of Secs. 2.3 and 4.

## 2 A PAC-Bayesian bound for unbounded martingales

### 2.1 Main result

A line of work led by Seldin et al. (2011; 2012a;b) provided PAC-Bayes bounds for almost surely bounded martingales. We provably extend the remits of their result to the case of unbounded martingales.

**Framework** Our framework is close to the one of Seldin et al. (2012a): we assume having access to a countable dataset $S = (z_i)_{i \geq 1} \in$ with no restriction on the distribution of $S$ (in particular the $z_i$ can depend on each others). We denote for any $m$, $S_m := (z_i)_{i=1..m}$ the restriction of $S$ to its $m$ first points. $(\mathcal{F}_i)_{i \geq 0}$ is a filtration adapted to $S$. We denote for any $i \in \mathbb{N}$ $\mathbb{E}_{i-1}[.] := \mathbb{E}[. \mid \mathcal{F}_{i-1}]$. We also precise the space $\mathcal{H}$ to be an index (or a hypothesis) space, possibly uncountably infinite. Let $\{X_1(S_1, h), X_2(S_2, h), \cdots : h \in \mathcal{H}\}$ be martingale difference sequences, meaning that for any $m \geq 1, h \in \mathcal{H}$, $\mathbb{E}_{m-1}[X_m(S_m, h)] = 0$.

For any $h \in \mathcal{H}$, let $M_m(h) = \sum_{i=1}^{m} X_i(S_i, h)$ be martingales corresponding to the martingale difference sequences and we define, as in Bercu and Touati (2008), the following

$$[M]_m(h) := \sum_{i=1}^{m} X_i(S_i, h)^2,$$

$$\langle M \rangle_m(h) = \sum_{i=1}^{m} \mathbb{E}_{i-1}\left[X_i(S_i, h)^2\right].$$

For a distribution $Q$ over $\mathcal{H}$ define weighted averages of the martingales with respect to $Q$ as $M_m(Q) = \mathbb{E}_{h \sim Q}[M_m(h)]$ (similar definitions hold for $[M]_m(Q), \langle M \rangle_m(Q)$).

**Main result.** We now present the main result of this section where we succesfully avoid the boundedness assumption on martingales. This relaxation comes at the cost of additional variance terms $[M]_m, \langle M \rangle_m$.

**Theorem 2.1.** *For any data-free prior $P \in \mathcal{M}_1^+(\mathcal{H})$, any $\lambda > 0$, any collection of martingales $(M_m(h))_{m \geq 1}$ indexed by $h \in \mathcal{H}$, the following holds with probability $1 - \delta$ over the sample $S = (z_i)_{i \in \mathbb{N}}$, for all $m \in \mathbb{N}/\{0\}$, $Q \in \mathcal{M}_1^+(\mathcal{H})$:*

$$|M_m(Q)| \leq \frac{\mathrm{KL}(Q, P) + \log(2/\delta)}{\lambda} + \frac{\lambda}{2}\left([M]_m(Q) + \langle M \rangle_m(Q)\right).$$

Proof lies in Sec. 2.2.

**Analysis of the bound.** This theorem involves several terms. The change of measure inequality introduces the KL divergence term, the approximation term $\log(2/\delta)$ comes from Ville's inequality (instead of Markov in classical PAC-Bayes). Finally, the terms $[M]_m(Q), \langle M \rangle_m(Q)$ come from our choice of supermartingale as suggested by Bercu and Touati (2008). The term $[M]_m(Q)$ can be interpreted as an empirical variance term while $\langle M \rangle_m(Q)$ is its theoretical counterpart. Note that $\langle M \rangle_m(Q)$ also appears in Seldin et al. (2012a, Theorem 1).

We recall that this general result stands with no assumption on the martingale difference sequence $(X_i)_{i \geq 1}$ and holds uniformly on all $m \geq 1$. Those two points are, to the best of our knowledge, new within the PAC-Bayes literature. We discuss in Sec. 2.3 and Appendix B more concrete instantiations.

**Comparison with literature.** The closest result from Thm. 2.1 is the PAC-Bayes Bernstein inequality of Seldin et al. (2012a). Our bound is a natural extension of theirs as their result only involves the variance term (not the empirical one), but requires two additional assumptions:

1. Bounded variations of the martingale difference sequence: $\forall m, \exists C_m \in \mathbb{R}^2$ such that a.s. for all $h$ $|X_m(S_m, h)| \leq C_m$.

2. Restriction on the range of the $\lambda$: $\forall m, \lambda_m \leq 1/C_m$.

Seldin et al. (2012a) need those assumptions to ensure the *Bernstein assumption* which states that for any $h$, $\mathbb{E}[\exp(\lambda M_m(h) - \frac{\lambda^2}{2}\langle M \rangle_m(h))] \leq 1$. Our proof technique do not require the Bernstein assumption (and so none of the two conditions described above, which allow us to deal with unbounded martingales) as we exploit the supermartingale structure to obtain our results. More precisely, the price to pay to avoid the Bernstein assumption is to consider the empirical variance term $[M]_m(h)$ and to prove that $\left(\exp\left(\lambda M_m - \frac{\lambda^2}{2}\left([M]_m + \langle M \rangle_m\right)\right)\right)_{m \geq 1}$ is a supermartingale using Lemma 1.2 and Lemma 1.3 (see Sec. 2.2 for the complete proof). A broader discussion is detailed in Appendix B.

## 2.2 Proof of Thm. 2.1

*Proof.* We fix $\eta \in \mathbb{R}$ and we consider the function $f_m$ to be for all $(S, h)$:

$$f_m(S, h) := \eta M_m(h) - \frac{\eta^2}{2}\left([M]_m(h) + \langle M \rangle_m(h)\right)$$
$$= \sum_{i=1}^{m} \eta \Delta M_i(h) - \frac{\eta^2}{2}(\Delta[M]_i(h) + \Delta\langle M \rangle_i(h)),$$

where $\Delta M_i(h) = X_i(S_i, h)$, $\quad \Delta[M]_i(h) = X_i(S_i, h)^2$, $\quad \Delta\langle M \rangle_i(h) = \mathbb{E}_{i-1}\left[X_i(S_i, h)^2\right]$. For the sake of clarity, we dropped the dependency in $S$ of $M_m$. Note that, given the definition of $M_m$, $M_m(h)$ is $\mathcal{F}_m$ measurable for any fixed $h$.

Let $P$ a fixed data-free prior, we first apply the change of measure inequality to obtain $\forall m \in \mathbb{N}, \forall Q \in \mathcal{M}_1^+(\mathcal{H})$:

$$\mathbb{E}_{h \sim Q}[f_m(S, h)] \leq \mathrm{KL}(Q, P) + \log\left(\underbrace{\mathbb{E}_{h \sim P}\left[\exp(f_m(S, h))\right]}_{:=Z_m}\right),$$

with the convention $f_0 = 0$. We now have to show that $(Z_m)_m$ is a supermartingale with $Z_0 = 1$. To do so remark that for any $m$, because $P$ is data free one has the following result.

**Lemma 2.2.** *For any data-free prior $P$, any $\sigma$-algebra $\mathcal{F}$ belonging to the filtration $(\mathcal{F}_i)_{i \geq 0}$, any nonnegative function $f$ taking as argument the sample $S$ and a predictor $h$, one has almost surely:*

$$\mathbb{E}\left[\mathbb{E}_{h \sim P}[f(S, h)] \mid \mathcal{F}\right] = \mathbb{E}_{h \sim P}\left[\mathbb{E}[f(S, h) \mid \mathcal{F}]\right].$$

*Proof.* Let $A$ be a $\mathcal{F}$-measurable event. We want to show that

$$\mathbb{E}\left[\mathbb{E}_{h \sim P}[f(S, h)]\mathbb{1}_A\right] = \mathbb{E}\left[\mathbb{E}_{h \sim P}\left[\mathbb{E}[f(S, h) \mid \mathcal{F}]\right]\mathbb{1}_A\right],$$

where the first expectation in each term is taken over $S$. Note that it is possible to take this expectation thanks to the Kolomogorov's extension theorem (see *e.g.* Tao, 2011, Thm 2.4.4) which ensure the existence of a probability space for the discrete-time stochastic process $S = (z_i)_{i \geq 1}$.

Thus, this is enough to conclude that

$$\mathbb{E}\left[\mathbb{E}_{h \sim P}[f(S, h)] \mid \mathcal{F}\right] = \mathbb{E}_{h \sim P}\left[\mathbb{E}[f(S, h) \mid \mathcal{F}]\right],$$

by definition of the conditional expectation. To do so, notice that because $f(S,h)\mathbb{1}_A$ is a nonnegative function, and that $P$ is data-free, we can apply the classical Fubini-Tonelli theorem.

$$\mathbb{E}\left[\mathbb{E}_{h\sim P}[f(S,h)]\mathbb{1}_A\right] = \mathbb{E}_{h\sim P}\left[\mathbb{E}\left[f(S,h)\mathbb{1}_A\right]\right].$$

One now conditions by $\mathcal{F}$ and use the fact that $\mathbb{1}_A$ is $\mathcal{F}$-measurable:

$$= \mathbb{E}_{h\sim P}\left[\mathbb{E}\left[\mathbb{E}\left[f(S,h)\mid\mathcal{F}\right]\mathbb{1}_A\right]\right].$$

We finally re-apply Fubini-Tonelli to re-intervert the expectations:

$$= \mathbb{E}\left[\mathbb{E}_{h\sim P}\left[\mathbb{E}\left[f(S,h)\mid\mathcal{F}\right]\mathbb{1}_A\right]\right].$$

This concludes the proof of Lemma 2.2. $\qquad\square$

We then use Lemma 2.2 with $f=\exp(f_m)$ and $\mathcal{F}=\mathcal{F}_{m-1}$ to obtain:

$$\mathbb{E}_{m-1}[Z_m] = \mathbb{E}_{h\sim P}\left[\mathbb{E}_{m-1}[(\exp(f_m(S,h)))]\right]$$
$$= \mathbb{E}_{h\sim P}\left[\exp(f_{m-1}(S,h))\mathbb{E}_{m-1}\left[\exp(\eta\Delta M_m(h) - \frac{\eta^2}{2}(\Delta[M]_m(h) + \Delta\langle M\rangle_m(h)))\right]\right],$$

with $f_{m-1}(S,h) = \sum_{i=1}^{m-1}\eta(\Delta M_i(h)) - \frac{\eta^2}{2}(\Delta[M]_i(h) + \Delta\langle M\rangle_i(h))$. Using Lemma 1.3 ensures that for any $h$,

$$\mathbb{E}_{m-1}[\exp(\eta\Delta M_m(h) - \frac{\eta^2}{2}(\Delta[M]_m(h) + \Delta\langle M\rangle_m(h))] \leq 1,$$

thus we have

$$\mathbb{E}_{m-1}[Z_m] \leq \mathbb{E}_{h\sim P}\left[\exp(f_{m-1}(S,h))\right] = Z_{m-1}.$$

Thus $(Z_m)_m$ is a nonnegative supermartingale with $Z_0=1$. We can use Ville's inequality (Lemma 1.2) which states that
$$\mathbb{P}_S\left(\exists m\geq 1: Z_m > \delta^{-1}\right) \leq \delta.$$

Thus, with probability $1-\delta$ over $S$, for all $m\in\mathbb{N}, Z_m\leq 1/\delta$. We then have the following intermediary result. For all $P$ a data-free prior, $\eta\in\mathbb{R}$, with probability $1-\delta$ over $S$, for all $m>0, Q\in\mathcal{M}_1^+(\mathcal{H})$

$$\eta M_m(Q) \leq \mathrm{KL}(Q,P) + \log(1/\delta) + \frac{\eta^2}{2}\left([M]_m(Q) + \langle M\rangle_m(Q)\right], \tag{3}$$

recalling that $M_m(Q) = \mathbb{E}_{h\sim Q}[M_m(h)]$, and that similar definitons hold for $[M]_m(Q), \langle M\rangle_m(Q)$. Thus, applying the bound with $\eta=\pm\lambda$ ($\lambda>0$) and taking an union bound gives, with probability $1-\delta$ over $S$, for any $m\in\mathbb{N}, Q\in\mathcal{M}_1^+(\mathcal{H})$

$$\lambda|M_m(Q)| \leq \mathrm{KL}(Q,P) + \log(2/\delta) + \frac{\lambda^2}{2}\left([M]_m(Q) + \langle M\rangle_m(Q)\right].$$

Dividing by $\lambda$ concludes the proof. $\qquad\square$

## 2.3 A corollary: Batch learning with iid data and unbounded losses

In this section, we instantiate Thm. 2.1 onto a learning theory framework with iid data. We show that our bound encompasses several results of literature as particular cases.

**Framework** We consider a *learning problem* specified by a tuple $(\mathcal{H},\mathcal{Z},\ell)$ consisting of a set $\mathcal{H}$ of predictors, the data space $\mathcal{Z}$, and a loss function $\ell:\mathcal{H}\times\mathcal{Z}\to\mathbb{R}^+$. We consider a countable dataset $S=(z_i)_{i\geq 1}\in\mathcal{Z}^{\mathbb{N}}$ and assume that sequence is iid following the distribution $\mu$. We also denote by $\mathcal{M}_1^+(\mathcal{H})$ is the set of probabilities on $\mathcal{H}$.

**Definitions** The *generalisation error $R$* of a predictor $h \in \mathcal{H}$ is $\forall h, R(h) = \mathbb{E}_{z \sim \mu}[\ell(h, z)]$, the *empirical error* of $h$ is $\forall h, R_m(h) = \frac{1}{m} \sum_{i=1}^{m} \ell(h, z_i)]$ and finally the *quadratic generalisation error $V$* of $h$ is $\forall h, Quad(h) = \mathbb{E}_{z \sim \mu}[\ell(h, z)^2]$. We also denote by *generalisation gap* for any $h$ the quantity $|R(h) - R_m(h)|$.

**Main result.** We now state the main result of this section. This bound is a corollary of Thm. 2.1 and fills the gap with learning theory.

**Theorem 2.3.** *For any data-free prior $P \in \mathcal{M}_1^+(\mathcal{H})$, any $\lambda > 0$ the following holds with probability $1 - \delta$ over the sample $S = (z_i)_{i \in \mathbb{N}}$, for all $m \in \mathbb{N}/\{0\}$, $Q \in \mathcal{M}_1^+(\mathcal{H})$*

$$\mathbb{E}_{h \sim Q}[R(h)] \leq \mathbb{E}_{h \sim Q}\left[ R_m(h) + \frac{\lambda}{2m} \sum_{i=1}^{m} \ell(h, z_i)^2 \right] + \frac{\mathrm{KL}(Q, P) + \log(2/\delta)}{\lambda m} + \frac{\lambda}{2} \mathbb{E}_{h \sim Q}[\mathrm{Quad}(h)].$$

Proof is furnished in Appendix C.

**About the choice of $\lambda$.** A novelty in this theorem is that the bound holds *simultaneously on all $m > 0$* – this is due to the use of Ville's inequality. This sheds a new light on the choice of $\lambda$. Indeed, taking a localised $\lambda$ depending on a given sample size (e.g. $\lambda_m = 1/\sqrt{m}$) ensures convergence guarantees for the expected generalisation gap. Doing so, our bound matches the usual PAC-Bayes literature (i.e. a bound holding with high probability for a single $m$). However the novelty brought by Thm. 2.3 is that our bound holds for unbounded losses for all times simultaneously. This suggests that taking a sample size-dependent $\lambda$ may not be the best answer. We detail an instance of this fact below when one thinks of $\lambda$ as a parameter of an optimisation objective. Indeed, our bound suggests a new optimisation objective for unbounded losses which is for any $m > 0$:

$$\mathrm{argmin}_Q \, \mathbb{E}_{h \sim Q}\left[ \frac{1}{m} \sum_{i=1}^{m} \left( \ell(h, z_i) + \frac{\lambda}{2} \ell(h, z_i)^2 \right) \right] + \frac{\mathrm{KL}(Q, P)}{\lambda m}. \tag{4}$$

Eq. (4) differs from the classical objective of Catoni (2007, Thm 1.2.6) on the additional quadratic term $\frac{\lambda}{2}\ell(h, z_i)^2$. Note that this objective implies a bound on the theoretical order 2 moment to be meaningful as we do not include it in our objective. Note that this constraint is less restrictive than Catoni's objective which requires a bounded loss. This objective stresses the role of the parameter $\lambda$ as being involved in a new explicit tradeoff between the KL term and the efficiency on training data.

Also, this optimisation objective is valid for any sample size $m$, this means that our $\lambda$ should not depend on certain dataset size but should be fixed in order to ensure a learning algorithm with generalisation guarantees at all time. This draws a parallel with Stochastic Gradient Descent with fixed learning step.

**About the underlying assumptions in this bound.** Our result is empirical (all terms can be computer or approximated) at the exception of the term $\mathbb{E}_{h \sim Q}[\mathrm{Quad}(h)]$. This invites to choose carefully the class of posteriors, in order to bound this second-order moment with minimal assumptions. For instance, if we consider the particular case of the quadratic loss $\ell(h, z) = (h - z)^2$, then we only need to assume that our data have a finite variance if we restrict our posteriors to have both bounded means and variance. This assumption is striclty less restrictive than the classical subgaussian/subgamma assumption classically appearing in the literature.

**Comparison with literature.** Back to the bounded case, we note that instantiating the boundedness assumption in Thm. 2.3 make us recover the result of Alquier et al. (2016, Theorem 4.1) for the subgaussian case. We also remark that instantiating the HYPE condition conditioning Haddouche et al. (2021, Theorem 3) allow us to improve their result as we transformed the control of an exponential moment into one on a second-order moment. More details are gathered in Appendix B. We also compare Thm. 2.3 to Kuzborskij and Szepesvári (2019, Theorem 3) which is a PAC-Bayes bound for unbounded losses obtained through a concentration inequality from De la Peña et al. (2009). They arrived to what they denote as semi-empirical inequalities which also involve empirical and theoretical variance terms (and not an exponential moment).Their bound holds for independent data and a single posterior. First of all, note that Thm. 2.3 holds for any posterior, which is strictly more general. Note also that our bound is a straightforward corollary

of Thm. 2.1 which holds for any martingale (thus for any data distribution in a learning theory framework) and so, exploits a different toolbox than Kuzborskij and Szepesvári (2019) (control of a supermartingale vs. concentration bounds for independent data). We insist that a fundamental novelty in our work is to extend the conclusion of Kuzborskij and Szepesvári (2019) to the case of non-independent data: it is possible to perform PAC-Bayes learning for unbounded losses at the expense of the control of second-order moments. Note also that their bound is slightly tighter than ours as their result is Thm. 2.3 being optimised in $\lambda$ (which is something we cannot do as the resulting $\lambda$ would be data-dependent).

## 3 Online PAC-Bayes learning with unbounded losses

Recently, an online learning framework has been designed in Haddouche and Guedj (2022). This allowed the design of Online PAC-Bayes (OPB) algorithms which involved the use of history-dependent priors evolving at each time step of the learning procedure. The main contribution of this section is an OPB bound valid for unbounded losses.

**Framework** We consider the same framework as in Sec. 2.3 except we do not make any assumption on the data distribution. Our goal is now to define a posterior sequence $(Q_i)_{i\geq 1}$ from a prior sequence $(P_i)_{i\geq 1}$. We also define a filtration $(\mathcal{F}_i)_{i\geq 1}$ adapted to $(z_i)_{i\geq 1}$. We reuse the following definitions extracted from Haddouche and Guedj (2022).

**Definitions** For all $i$, we denote by $\mathbb{E}_i[.]$ the conditional expectation $\mathbb{E}[. \mid \mathcal{F}_i]$.

A *stochastic kernel* from $\cup_{m=1}^{\infty} \mathcal{Z}^m$ to $\mathcal{H}$ is defined as a mapping $Q : \cup_{m=1}^{\infty} \mathcal{Z}^m \times \Sigma_{\mathcal{H}} \to [0,1]$ where (i) For any $B \in \Sigma_{\mathcal{H}}$, the function $S \mapsto Q(S,B)$ is measurable, (ii) For any $S$, the function $B \mapsto Q(S,B)$ is a probability measure over $\mathcal{H}$.

We also say that a sequence of stochastic kernels $(P_i)_{i\geq 1}$ is an *online predictive sequence* if (i) for all $i \geq 1, S \in \cup_{m=1}^{\infty} \mathcal{Z}^m, P_i(S,.)$ is $\mathcal{F}_{i-1}$ measurable and (ii) for all $i \geq 2, P_i(S,.) \gg P_1(S,.)$.

**Main result.** We now state the main theorem of this section, which extends the remits of the Online PAC-Bayes framework to the case of unbounded losses.

**Theorem 3.1.** *For any distribution over the (countable) dataset $S$, any $\lambda > 0$ and any online predictive sequence (used as priors) $(P_i)_{i\geq 1}$, we have with probability at least $1-\delta$ over the sample $S \sim \mu$, the following, holding for the data-dependent measures $P_{i,S} := P_i(S,.)$ any posterior sequence $(Q_i)_{i\geq 1}$ and any $m \geq 1$:*

$$\sum_{i=1}^{m} \mathbb{E}_{h_i \sim Q_i} \left[ \mathbb{E}[\ell(h_i, z_i) \mid \mathcal{F}_{i-1}] \right] \leq \sum_{i=1}^{m} \mathbb{E}_{h_i \sim Q_i} \left[ \ell(h_i, z_i) \right] + \frac{\lambda}{2} \sum_{i=1}^{m} \mathbb{E}_{h_i \sim Q_i} \left[ \hat{V}_i(h_i, z_i) + V_i(h_i) \right]$$

$$+ \sum_{i=1}^{m} \frac{\mathrm{KL}(Q_i \| P_{i,S})}{\lambda} + \frac{\log(1/\delta)}{\lambda}.$$

*With for all $i$, $\hat{V}_i(h_i, z_i) = (\ell(h_i, z_i) - \mathbb{E}_{i-1}[\ell(h_i, z_i)])^2$ is the empirical variance at time $i$ and $V_i(h_i) = \mathbb{E}_{i-1}[\hat{V}(h_i, z_i)]$ is the true conditional variance.*

Proof lies in Sec. 3.1.

**Analysis of the bound.** This bound is, to our knowledge, the first Online PAC-Bayes bound in literature holding for unbounded losses. It is semi-empirical as the variance and empirical variance terms have theoretical components. However, these terms can be controlled with assumptions on conditional second-order moments and not on exponential ones (as made in Haddouche and Guedj, 2022 where the bounded loss assumption was used to obtain conditional subgaussianity). To emphasise our point, we consider as in Sec. 2.3 the case of the quadratic loss $\ell(h, z) = (h - z)^2$. Here, we only need to assume that our data have a finite variance if we restrict our posteriors to have both bounded means and variance. Also the meaning of the online predictive sequence $P_i$ is that we must be able to design properly a sequence of priors before drawing

our data, this can be for instance an online algorithm whihc generate a prior distribution from past data at each time step.

Finally, we note that if we assume being able to bound simultaneaously all condtional means and variance (which is strictly less restrictive than bounding the loss),then Thm. 3.1 suggests a new online learning objective which is an online counterpart to Eq. (4).

$$\forall i \geq 1 \ \hat{Q}_{i+1} = \underset{Q \in \mathcal{M}_1^+(\mathcal{H})}{\operatorname{argmin}} \ \mathbb{E}_{h_i \sim Q} \left[ \ell(h_i, z_i) + \frac{\lambda}{2} \ell(h_i, z_i)^2 \right] + \frac{\mathrm{KL}(Q \| P_{i,S})}{\lambda} \tag{5}$$

**Comparison with literature.** Our most natural comparison point is Theorem 2.3 of Haddouche and Guedj (2022) (re-stated in Appendix A). We claim that Thm. 3.1 is a strict improvement of their result on various sides described below.

- If we assume our loss to be bounded, then we can upper bound our empirical/theoretical variance terms to recover exactly Haddouche and Guedj (2022, Theorem 2.3). Our bound can then be seen as a strict extension of theirs and shows that bounding order two moments is a sufficient condition to perform online PAC-Bayes: subgaussianity induced by boundedness is not necessary even when our data are non iid.

- Another crucial point lies on the range of our result which holds with high probability for any countable posterior sequence $(Q_i)_{i \geq 1}$, any time $m$ and the priors $(P_{i,S})_{i \geq 1}$. This is far much general than Haddouche and Guedj (2022, Theorem 2.3) which holds only for a single $m$ and a single posterior sequence $(Q_{i,S})_{i=1..m}$. This happens because in Haddouche and Guedj (2022), the change of measure inequality has not been exploited: they used a preliminary theorem from Rivasplata et al. (2020) which holds for a single (data-dependent) prior/posterior couple. This preliminary theorem already involved Markov's inequality which forced the authors to assume conditionnal subgaussianity to deal with an exponential moment. On the contrary, we exploited the fact that our online predictive sequence was history-dependent to use the change of measure inequality at any time step and control an exponential supermartingale through Ville's inequality.

- In Haddouche and Guedj (2022, Eq. 1), an OPB algorithm is given by their upper bound. This works because their associated learning objective admits a close form (Gibbs posterior) which matches the fact their bound hold for a single posterior sequence. Because our bound holds uniformly on all posteriors, it is now legitimate to restrict their algorithms to any parametric class of distributions and perform any optimisation algorithm to obtain a surrogate of the best candidate.

Online PAC-Bayes as presented in Haddouche and Guedj (2022) relies on a conditional subgaussiannity assumption to control an exponential moment. They did not exploit a martingale-type structure to do so. Our supermartingale approach has proven to be well suited to Online PAC-Bayes as we provided atheorem valid for unbounded losses holding simultaneously on all posteriors: two points which have not been reached in Haddouche and Guedj (2022).

### 3.1 Proof of Thm. 3.1

*Proof.* We fix $m \geq 1$, $S$ a countable dataset and $(P_i)_{i \geq 1}$ an online predictive sequence. We aim to design a $m$-tuple of probabilities. Thus, our predictor set of interest is $\mathcal{H}_m := \mathcal{H}^{\otimes m}$ and then, our predictor $h$ is a tuple $(h_1, .., h_m) \in \mathcal{H}$.

Our goal is to apply the change of measure inequality on $\mathcal{H}_m$ to a specific function $f_m$ inspired from Lemma 1.3. We define this function below, for any sample $S$ and any predictor $h^m = (h_1, ..., h_m)$

$$f_m(S, h^m) := \sum_{i=1}^m \lambda X_i(h_i, z_i) - \frac{\lambda^2}{2} \sum_{i=1}^m (\hat{V}_i(h_i, z_i) + V_i(h_i)),$$

where $X_i(h_i, z_i) = \mathbb{E}_{i-1}[\ell(h_i, z_i)] - \ell(h_i, z_i)$. Notice that for fixed $h$, the sequence $(f_m(S, h))_{m \geq 1}$ is a super-martingale according to Lemma 1.3.

Now for a given posterior tuple $Q_1, ... Q_m$ we define $Q = Q_1 \otimes ... \otimes Q_m$ and also $P_S^m = P_{1,S} \otimes ... \otimes P_{m,S}$. We can now properly apply the change of measure inequality for any $m$:

$$\sum_{i=1}^m \mathbb{E}_{h_i \sim Q_i}[\lambda X_i(h_i, z_i) - \frac{\lambda^2}{2}(\hat{V}_i(h_i, z_i) + V_i(h_i))] = \mathbb{E}_{h^m \sim Q}[f_m(S, h^m)]$$
$$\leq \mathrm{KL}(Q, P_S^m) + \log\left(\mathbb{E}_{h^m \sim P_S^m} \exp(f_m(S, h^m))\right).$$

Noticing that $\mathrm{KL}(Q, P_S^m) = \sum_{i=1}^m \mathrm{KL}(Q_i, P_{i,S})$, the only remaining term to deal with is the exponential rv. To do so we prove the following lemma:

**Lemma 3.2.** *The sequence* $(M_m := \mathbb{E}_{h^m \sim P_S^m} \exp(f_m(S, h^m)))_{m \geq 1}$ *is a non-negative supermartingale.*

*Proof.* We fix $m \geq 1$ and we recall that for any $i$, $P_{i,S}$ is $\mathcal{F}_{i-1}$-measurable. We show that $\mathbb{E}_{m-1}[M_m] \leq M_{m-1}$. We first recover $M_{m-1}$ from $\mathbb{E}_{m-1}[M_m]$.

$$\begin{aligned}
\mathbb{E}_{m-1}[M_m] &= \mathbb{E}_{m-1}\left[\mathbb{E}_{h^m \sim P_S^m} \exp(f_m(S, h^m))\right] \\
&= \mathbb{E}_{m-1}\left[\mathbb{E}_{h_1,..,h_m \sim P_{1,S} \otimes ... \otimes P_{m,S}} \exp(f_m(S, h^m))\right] \\
&= \mathbb{E}_{m-1}\left[\mathbb{E}_{h_1,..,h_m \sim P_{1,S} \otimes ... \otimes P_{m,S}} \left[\Pi_{i=1}^m \exp\left(\lambda X_i(h_i, z_i) - \frac{\lambda^2}{2}(\hat{V}_i(h_i, z_i) + V_i(h_i))\right)\right]\right] \\
&= M_{m-1}\mathbb{E}_{m-1}\left[\mathbb{E}_{h_m \sim P_{m,S}}\left[\exp\left(\lambda X_m(h_m, z_m) - \frac{\lambda^2}{2}(\hat{V}_m(h_m, z_m) + V_m(h_m))\right)\right]\right].
\end{aligned}$$

The last line holding because $P_S^{m-1} = P_{1,S} \otimes ... \otimes P_{m-1,S}$ is $\mathcal{F}_{m-1}$ measurable.

Now we exploit the fact that $P_{m,S}$ is $\mathcal{F}_{m-1}$ measurable to apply a conditional Fubini lemma stated in Haddouche and Guedj (2022, Lemma D.3). We have:

$$\mathbb{E}_{m-1}\left[\mathbb{E}_{h_m \sim P_{m,S}}\left[\exp\left(\lambda X_m(h_m, z_m) - \frac{\lambda^2}{2}(\hat{V}_m(h_m, z_m) + V_m(h_m))\right)\right]\right]$$
$$= \mathbb{E}_{h_m \sim P_{m,S}}\left[\mathbb{E}_{m-1}\left[\exp\left(\lambda X_m(h_m, z_m) - \frac{\lambda^2}{2}(\hat{V}_m(h_m, z_m) + V_m(h_m))\right)\right]\right].$$

Now we can apply Lemma 1.3 for any $h_m \in \mathcal{H}$ with $\Delta M_m = X_m(h_m, z_m), \Delta[M]_m = \hat{V}(h_m, z_m)$ and $\Delta\langle M\rangle_m = V_m(h_m)$. We then have for all $h_m \in \mathcal{H}$:

$$\mathbb{E}_{m-1}\left[\exp\left(\lambda X_m(h_m, z_m) - \frac{\lambda^2}{2}(\hat{V}_m(h_m, z_m) + V_m(h_m))\right)\right] \leq 1.$$

Thus $\mathbb{E}_{m-1}[M_m] \leq M_{m-1}$, this concludes the lemma's proof. $\qquad\square$

Now we can apply Ville's inequality which implies that with probability at least $1 - \delta$, for any $m \geq 1$:

$$\mathbb{E}_{h^m \sim P_S^m} \exp(f_m(S, h^m)) \leq \frac{1}{\delta}.$$

Thus we have with probability at least $1 - \delta$, for any posterior sequence $(Q_i)_{i \geq 1}$, the data-dependent measures $P_{1,S}, ..., P_{m,S}$ and any $m \geq 1$:

$$\sum_{i=1}^{m} \mathbb{E}_{h_i \sim Q_i} \left[ \lambda X_i(h_i, z_i) - \frac{\lambda^2}{2}(\hat{V}_i(h_i, z_i) + V_i(h_i)) \right] \leq \sum_{i=1}^{m} \mathrm{KL}(Q_i, P_{i,S}) + \log\left(\frac{1}{\delta}\right).$$

Re-organising the terms in this bound and dividing by $\lambda$ concludes the proof.

$\square$

## 4  Application to the multi-armed bandit problem

We exploit our main result in the context of the multi-armed bandit problem – we adopt the framework of Seldin et al. (2012a).

**Framework.** Let $\mathcal{A}$ be a set of actions of size $|\mathcal{A}| = K < +\infty$ and $a \in \mathcal{A}$ be an action. At each round $i$, the environment furnishes a reward function $R_i : \mathcal{A} \to \mathbb{R}$ which associate a reward $R_i(a)$ to the arm $a$. Assuming the $R_i$s are iid, we denote for any $a$, the *expected reward for action $a$* to be $R(a) = \mathbb{E}_{R_1}[R_1(a)]$. At each round $i$, the player executes an action $A_i$ according to a policy $\pi_i$. We then set the filtration $(\mathcal{F}_i)_{i \geq 1}$ to be $\mathcal{F}_i = \sigma(\{\pi_j, A_j, R_j \mid 1 \leq j \leq m\})$.

**Assumptions.** We suppose here that $(R_i)_{i \geq 1}$ is an iid sequence and that at each time $i$, $A_i$ and $R_i$ are independent and that $\pi_i$ is $\mathcal{F}_{i-1}$ measurable. This means that the player is not aware of the rewards each round and performs its current move with regards to the past.

We also add two technical assumptions. First, the order two moment of the expected reward is uniformly bounded: $\sup_{a \in \mathcal{A}} \mathbb{E}_{R_1}[R_1(a)^2] \leq C$. This assumption is strictly less restrictive than the boundedness assumption made in Seldin et al. (2012a). Similarly to this work, we also assume that there exists a sequence $(\varepsilon_i)_{i \geq 1}$ such that $\inf_{a \in \mathcal{A}} \pi_i(a) \geq \varepsilon_i$. We say that $(\pi_i)_{i \geq 1}$ is *bounded from below by $(\varepsilon_i)_{i \geq 1}$*.

**Definitions.** For $i \geq 1$ and $a \in \{1, \ldots, K\}$, define a set of random variables $(R_i^a)_{i \geq 1}$ (*the importance weighted samples*, Sutton and Barto, 2018)

$$R_i^a := \begin{cases} \frac{1}{\pi_i(a)} R_i, & \text{if } A_i = a, \\ 0, & \text{otherwise.} \end{cases}$$

We define for any time $m$: $\hat{R}_m(a) = \frac{1}{m} \sum_{i=1}^{t} R_i^a$. Observe that for all $i$, $\mathbb{E}[R_i^a \mid \mathcal{F}_{i-1}] = R(a)$ and $\mathbb{E}[\hat{R}_m(a)] = R(a)$. Let $a^*$ be the "best" action (the action with the highest expected reward, if there are multiple "best" actions pick any of them). Define the *expected and empirical per-round regrets* as

$$\Delta(a) = R(a^*) - R(a), \quad \hat{\Delta}_m(a) = \hat{R}_m(a^*) - \hat{R}_m(a).$$

Observe that $m\left(\hat{\Delta}_m(a) - \Delta(a)\right)$ forms a martingale. Let

$$V_m(a) = \sum_{i=1}^{m} \mathbb{E}\left[ \left( R_i^{a^*} - R_i^a - [R(a^*) - R(a)] \right)^2 \mid \mathcal{F}_{i-1} \right]$$

be the cumulative variance of this martingale and

$$\hat{V}_m(a) = \sum_{i=1}^{m} \left( R_i^{a^*} - R_i^a - [R(a^*) - R(a)] \right)^2$$

its empirical counterpart. We denote for any distribution $Q$ over $\mathcal{A}$, $\Delta(Q) = \mathbb{E}_{a \sim Q}[\Delta(a)]$, $V_m(Q) = \mathbb{E}_{a \sim Q}[V_m(a)]$, similar definitions hold for $\hat{\Delta}_m(Q), \hat{V}_m(Q)$. We can now state the main result of this section – its proof is deferred to Appendix C.

**Theorem 4.1.** *For any $m \geq 1$, any history-dependent policy sequence $(\pi_i)_{i \geq 1}$ bounded from below by $(\varepsilon_i)_{i \geq 1}$, we have with probability $1 - \delta$, for all posterior $Q$*

$$\left| \Delta(Q) - \hat{\Delta}_m(Q) \right| \leq 2 \sqrt{\frac{\left(1 + \frac{2K}{\delta}\right)\left(\log(K) + \log(4/\delta)\right)}{m \varepsilon_m}}.$$

To the best of our knowledge, this result is the first PAC-Bayesian guarantees for multi-armed bandits with unbounded rewards. The proposed bound is as tight as Theorem 2.3 of Seldin et al. (2012a), up to a factor $(e - 2)$ transformed into $\left(1 + \frac{2K}{\delta}\right)$ (which is a huge dependency in $K$) within the square root. Note that our result comes at the price of the localisation: Theorem 2.3 of Seldin et al. (2012a) proposes a bound holding uniformly for all time $m$ while our approach only holds for a single time $m$.

We believe there is room for improvement in Thm. 4.1. Indeed, the current approach is naive as it consists in bounding crudely with high probability the empirical variance. Such a naive trick impeach us to consider all times simultaneously. Indeed, in its current form, taking an union bound on Thm. 4.1 is costful as we have a dependency in $1/\delta$ in our result (instead of $\log(1/\delta)$ in Seldin et al., 2012a): this would destroy the convergence rate. The question of dealing more subtly with the empirical variance term is left as an open question.

## 5 Conclusion

We showed that it is possible to generalise the PAC-Bayes toolbox to unbounded martingales and heavy-tailed losses (resp. learning problem with unbounded losses for batch/online learning), the solely implicit assumption being the existence of second order moments on the martingale difference sequence (resp. on the loss function) which is reasonable as many PAC-Bayes bound lies on assumptions on exponential moments (*e.g.* the subgaussian assumption) to work. We also proved that our main theorem can be seen as a general basis allowing to recover several PAC-Bayesian bounds. This shows that the supermartingale framework is a fruitful approach to unify several branches of PAC-Bayes and could lead to new promising developement such as the work of Jang et al. (2023).

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

# A  Some PAC-Bayesian background

We present below an immediate corollary of Seldin et al. (2012a, Thm 2.1) where we upper bounded the cumulative by an empirical quantity (the sum of squared upper bound of the martingale difference sequence).

**Theorem A.1** (Seldin et al., 2012a, Theorem 2.1). *Let $\{C_1, C_2, \ldots\}$ be an increasing sequence set in advance, such that $|X_i(S_i, h)| \leq C_i$ for all $S_i, h$ with probability 1. Let $\{P_1, P_2, \ldots\}$ be a sequence of data-free prior distributions over $\mathcal{H}$. Let $(\lambda_i)_{i \geq 1}$ be a sequence of positive numbers such that*

$$\lambda_m \leq \frac{1}{C_m}.$$

*Then with probability $1 - \delta$ over $S = (z_i)_{i \geq 1}$, for all $m \geq 1$, any posterior $Q$ over $\mathcal{H}$,*

$$|M_m(Q)| \leq \frac{KL(Q\|P_m) + 2\log(m+1) + \log\frac{2}{\delta}}{\lambda_m} + (e - 2)\lambda_m V_m(Q),$$

*where $V_m(Q)$ is defined in Appendix B.1.*

*Furthermore, if we bound the variance term, we would have:*

$$|M_m(Q)| \leq \frac{KL(Q\|P_m) + 2\log(m+1) + \log\frac{2}{\delta}}{\lambda_m} + (e - 2)\lambda_m \sum_{i=1}^{m} C_i^2.$$

Below, we use the definitions introduced in Sec. 2.3. We study here a particular case of Alquier et al. (2016) for bounded losses which are especially subgaussian thanks to Hoeffding's lemma.

**Theorem A.2** (Adapted from Alquier et al., 2016, Theorem 4.1). *Let $m > 0, S = (z_1, ..., z_m)$ be an iid sample from the same law $\mu$. For any data-free prior $P$, for any loss function $\ell$ bounded by $K$, any $\lambda > 0, \delta \in ]0; 1[$, one has with probability $1 - \delta$ for any posterior $Q \in \mathcal{M}_1(\mathcal{H})$*

$$\mathbb{E}_{h \sim Q}[R(h)] \leq \mathbb{E}_{h \sim Q}[R_m(h)] + \frac{KL(Q\|P) + \log(1/\delta)}{\lambda} + \frac{\lambda K^2}{2m}.$$

**Theorem A.3** (Haddouche et al., 2021, Theorem 3). *Let the loss $\ell$ be $\mathrm{HYPE}(K)$ compliant. For any $P \in \mathcal{M}_1^+(\mathcal{H})$ with no data dependency, for any $\alpha \in \mathbb{R}$ and for any $\delta \in [0, 1]$, we have with probability at least $1 - \delta$ over size-$m$ samples $S$, for any $Q$*

$$\mathbb{E}_{h \sim Q}[R(h)] \leq \mathbb{E}_{h \sim Q}[R_m(h)] + \frac{KL(Q\|P) + \log\left(\frac{1}{\delta}\right)}{m^\alpha} + \frac{1}{m^\alpha} \log\left(\mathbb{E}_{h \sim P}\left[\exp\left(\frac{K(h)^2}{2m^{1-2\alpha}}\right)\right]\right).$$

**Theorem A.4** (Theorem 2.3 of Haddouche and Guedj, 2022). *For any distribution $\mu$ over $\mathcal{Z}^m$, any $\lambda > 0$ and any online predictive sequence (used as priors) $(P_i)$, for any sequence of stochastic kernels $(Q_i)$ we have with probability $1 - \delta$ over the sample $S \sim \mu$, the following, holding for the data-dependent measures $Q_{i,S} := Q_i(S, .), P_{i,S} := P_i(S, .)$ :*

$$\sum_{i=1}^{m} \mathbb{E}_{h_i \sim Q_{i,S}}[\mathbb{E}[\ell(h_i, z_i) \mid \mathcal{F}_{i-1}]] \leq \sum_{i=1}^{m} \mathbb{E}_{h_i \sim Q_{i,S}}[\ell(h_i, z_i)] + \frac{KL(Q_{i,S}\|P_{i,S})}{\lambda} + \frac{\lambda m K^2}{2} + \frac{\log(1/\delta)}{\lambda}.$$

# B  Extensions of previous results

Here we gather several corollaries of our main result in order to show how our Thm. 2.1 extends the validity of some classical results in the literature. More precisely we show that our result extends (up to numerical factors) the PAC-Bayes Bernstein inequality of Seldin et al. (2012a). Then, going back to the bounded case, we generalise a result from Catoni (2007) reformulated in Alquier et al. (2016) and we also show how our work strictly improves on the bound of Haddouche et al. (2021).

## B.1 Extension of the PAC-Bayes Bernstein inequality

Here we rename two terms for consistency with Theorem 2.1 of Seldin et al. (2012a) (see Thm. A.1). For a martingale $M_m(h) = \sum_{i=1}^m X_i(S_i, h)$, we define, at time $m$, *empirical cumulative variance* to be $\hat{V}_m(h) = [M]_m(h) = \sum_{i=1}^m X_i(S_i, h)^2$ and the *cumulative variance* as $V_m(h) = \langle M \rangle_m(h) = \sum_{i=1}^m \mathbb{E}_{i-1}[X_i(S_i, h)^2]$.

We provide below a corollary containing two bounds: the first one being a straightforward corollary of Thm. 2.1, the second being valid for bounded martingales and formally close to Theorem 2.1 of Seldin et al. (2012a).

**Corollary B.1.** *Let $\{P_1, P_2, \ldots\}$ be a sequence of data-free prior distributions over $\mathcal{H}$. Let $(\lambda_i)_{i \geq 1}$ be a sequence of positive numbers. Then the following holds with probability $1 - \delta$ over $S = (z_i)_{i \geq 1}$: for any tuple $(m, \lambda_k, P_k)$ with $m, k \geq 1$, any posterior $Q$ over $\mathcal{H}$,*

$$|M_m(Q)| \leq \frac{KL(Q, P_k) + 2\log(k+1) + \log(2/\delta)}{\lambda_k} + \frac{\lambda_k}{2}\left(\hat{V}_m(Q) + V_m(Q)\right), \tag{6}$$

*with $\hat{V}_m(Q) = \mathbb{E}_{h \sim Q}[\hat{V}_m(h)], V_m(Q) = \mathbb{E}_{h \sim Q}[V_m(h)]$. Furthermore, if we assume that for any $i$, there exists $C_i > 0$ such that $|X_i(S_i, h)| \leq C_i$ for all $S_i, h$ then we have the following corollary: with probability $1 - \delta$ over $S$, for any tuple $(m, \lambda_m, P_m)$ $m \geq 1$, any posterior $Q$,*

$$|M_m(Q)| \leq \frac{KL(Q, P_m) + 2\log(m+1) + \log(2/\delta)}{\lambda_m} + \lambda_m \sum_{i=1}^m C_i^2. \tag{7}$$

The proof is deferred to Appendix C. Note that Eq. (6) holds uniformly on all tuples $\{(\lambda_k, P_k, m) \mid k \geq 1, m \geq 1\}$ while Eq. (7), as well as Theorem 2.1 of Seldin et al. (2012a) holds uniformly on the tuples $\{(\lambda_m, P_m, m) \mid m \geq 1\}$ which is a strictly smaller collection. Hence our approach gives guarantees for a larger event with the same confidence level.

Furthermore, Theorem 2.1 of Seldin et al. (2012a) involves the cumulative variance $V_m(Q)$ (and not its empirical counterpart). Because this term is theoretical, we bound it in Thm. A.1 by $\sum_{i=1}^m C_i^2$ which is supposedly empirical. In this context, Eq. (7), recovers nearly exactly the bound of Seldin et al. (2012a) with the transformation of a factor $(e - 2)$ into 1. Notice also that Eq. (7) stands with no assumption on the range of the $\lambda_i$, which is not the case in Thm. A.1.

Finally, we stress two fundamental differences between our work and the one of Seldin et al. (2012a). First, we replace Markov's inequality by Ville's inequality; second, we exploited the exponential inequality of Lemma 1.3 instead of the Bernstein inequality. These allow for results for unbounded martingales for all $m$ simultaneously.

## B.2 Extensions of learning theory results

### B.2.1 A general result for bounded losses

We use definitions from Sec. 2.3 and provide a corollary of our main result when the loss is bounded by a positive constant $K > 0$. We assume our data are iid.

**Corollary B.2.** *For any data-free prior $P \in \mathcal{M}_1^+(\mathcal{H})$, any $\lambda > 0$ the following holds with probability $1 - \delta$ over the sample $S = (z_i)_{i \in \mathbb{N}}$, for all $m \in \mathbb{N}/\{0\}$, $Q \in \mathcal{M}_1^+(\mathcal{H})$*

$$|\mathbb{E}_{h \sim Q}[R(h)] - \mathbb{E}_{h \sim Q}[R_m(h)]| \leq \frac{KL(Q, P) + \log(2/\delta)}{\lambda m} + \lambda K^2.$$

*We also have the local bound: for any $m \geq 1$, with probability $1 - \delta$ over $S$, for all $Q \in \mathcal{M}_1^+(\mathcal{H})$*

$$\mathbb{E}_{h \sim Q}[R(h)] \leq \mathbb{E}_{h \sim Q}[R_m(h)] + \frac{KL(Q, P) + \log(2/\delta)}{\lambda} + \frac{\lambda K^2}{m}.$$

The proof is deferred to Appendix C. Remark that the second bound of Corollary B.2 is exactly the Catoni bound stated in Alquier et al. (2016) (see Thm. A.2 in Appendix A) up to a numerical factor of 2.

The first bound is, to our knowledge, the first PAC-Bayesian bound for bounded losses holding uniformly (for a given parameter $\lambda$) on the choice of $Q, m$ and thus extends the scope of Catoni's bound which holds for a single $m$ with high probability. Indeed, if we want for instance Thm. A.2 to hold for any $i \in \{1..m\}$, we then have to take an union bound on $m$ events which turns the term $\log(1/\delta)$ into $\log(m/\delta)$ (but with the benefit of holding for $m$ parameters $\lambda_1, ..., \lambda_m$). This point is common to the most classical PAC-Bayesian bounds (as those of McAllester, 1998; 1999; Maurer, 2004; Catoni, 2007; Tolstikhin and Seldin, 2013) and impeach us to have a bound uniformly on all $m \in \mathbb{N}/\{0\}$ as $\log(m)$ goes to infinity asymptotically.

### B.2.2 An extension of Haddouche et al. (2021)

We now focus on the work of Haddouche et al. (2021) which provides general PAC-Bayesian bounds for unbounded losses. Their theorems hold for iid data and under the so-called *HYPE* (for HYPothesis-dependent rangE) condition. It states that a loss function $\ell$ is $HYPE(K)$ compliant if there exists a function $K : \mathcal{H} \to \mathbb{R}^+$ (supposedly accessible) such that $\forall z \in \mathcal{Z}, \ell(h, z) \leq K(h)$. We provide Cor. B.3 to compare ourselves with their main result (stated in Thm. A.3 for convenience).

**Corollary B.3.** *For any data-free prior $P \in \mathcal{M}_1^+(\mathcal{H})$, any loss function $\ell$ being* HYPE($K$) *compliant, any $\alpha \in [0, 1], m \geq 1$, the following holds with probability $1 - \delta$ over the sample $S = (z_i)_{i \in \mathbb{N}}$, for all $Q \in \mathcal{M}_1^+(\mathcal{H})$*

$$\mathbb{E}_{h \sim Q}[R(h)] \leq \mathbb{E}_{h \sim Q} \left[ \frac{1}{m} \sum_{i=1}^{m} \left( \ell(h, z_i) + \frac{1}{2m^{1-\alpha}} \ell(h, z_i)^2 \right) \right] + \frac{\mathrm{KL}(Q, P) + \log(1/\delta)}{m^\alpha} + \frac{1}{2m^{1-\alpha}} \mathbb{E}_{h \sim Q}[K^2(h)].$$

*Proof.* The proof is a straightforward application of Thm. 2.3 by fixing $m \geq 1$ choosing $\lambda = m^{\alpha-1}$ (thus we localise Thm. 2.3 to a single $m$), and bounding $\mathrm{Quad}(h)$ by $K^2(h)$. $\square$

The main improvement of our bound over Thm. A.3 is that we do not have to assume the convergence of an exponential moment to obtain a non-trivial bound. Indeed, we transformed the (implicit) assumption $\mathbb{E}_{h \sim P} \left[ \exp \left( \frac{K(h)^2}{2m^{1-2\alpha}} \right) \right] < +\infty$ onto $\mathbb{E}_{h \sim Q}[K(h)^2] < +\infty$, which is significantly less restrictive. Furthermore, Thm. A.3 holds for a single choice of $m$ while ours still holds uniformly over all integers $m > 0$.

Cor. B.3 also sheds new light on the *HYPE* condition. Indeed, in Haddouche et al. (2021), $K$ only intervenes in an exponential moment involving the prior $P$, while ours considers a second-order moment on $K$ implying the posterior $Q$. The difference is major as $\mathbb{E}_{h \sim Q}[K(h)^2]$ can be controlled by a wise choice of posterior. Thus it can be incorporated in our optimisation route, acting now as an optimisation constraint instead of an environment constraint.

## C  Proofs

### C.1  Proof of Thm. 2.3

*Proof.* Let $P$ a fixed data-free prior, set $(\mathcal{F}_i)_{i \geq 0}$ such that for all $i$, $z_i$ is $\mathcal{F}_i$ measurable. We also set for any fixed $h \in \mathcal{H}, M_m(h) := \sum_{i=1}^{m} \ell(h, z_i) - R(h)$. Note that because data are iid, for any fixed $h$, the sequence $(M_m(h))_m$ is indeed a martingale. We set for any $m \geq 1, h \in \mathcal{H}$

$$[M]_m(h) = \sum_{i=1}^{m} (\ell(h, z_i) - R(h))^2$$

and

$$\langle M \rangle_m(h) = \sum_{i=1}^{m} \mathbb{E}_{i-1}[(\ell(h, z_i) - R(h))^2] = \sum_{i=1}^{m} \mathbb{E}_{z \sim \mu}[(\ell(h, z) - R(h))^2].$$

The last equality holds because data is assumed iid. Thus, we can apply Thm. 2.1 to obtain with probability $1 - \delta$

$$|M_m(Q)| \leq \frac{\mathrm{KL}(Q, P) + \log(2/\delta)}{\lambda} + \frac{\lambda}{2} \left([M]_m(Q)^2 + \langle M \rangle_m(Q)^2\right).$$

Now, we notice that $|M_m(Q)| = m |\mathbb{E}_{h \sim Q}[R(h) - R_m(h)]|$ and that for any $m, h$, because $\ell$ is nonnegative

$$[M]_m(h) + \langle M \rangle_m(h) = \sum_{i=1}^m (\ell(h, z_i) - R(h))^2 + \mathbb{E}_{z \sim \mu}[(\ell(h, z) - R(h))^2]$$

$$\leq \sum_{i=1}^m \ell(h, z_i)^2 + R(h)^2 + \mathbb{E}_{z \sim \mu}[\ell(h, z)^2] - R(h)^2.$$

Thus integrating over $h$ gives:

$$[M]_m(Q) + \langle M \rangle_m(Q) \leq \sum_{i=1}^m \mathbb{E}_{h \sim Q}[\ell(h, z_i)^2] + m \mathbb{E}_{h \sim Q}[\mathrm{Quad}(h)].$$

Then dividing by $m$ and applying the last inequality gives

$$\mathbb{E}_{h \sim Q}[R(h)] \leq \mathbb{E}_{h \sim Q}\left[\frac{1}{m} \sum_{i=1}^m \left(\ell(h, z_i) + \frac{\lambda}{2}\ell(h, z_i)^2\right)\right] + \frac{\mathrm{KL}(Q, P) + \log(2/\delta)}{\lambda m} + \frac{\lambda}{2} \mathbb{E}_{h \sim Q}[\mathrm{Quad}(h)].$$

This concludes the proof. $\qquad\square$

## C.2   Proof of Thm. 4.1

*Proof.* Let $(\lambda_m)_{i \geq 1}$ be a countable sequence of positive scalars. As precised earlier $M_m(a) := m\left(\hat{\Delta}_m(a) - \Delta(a)\right)$ is a martingale. We then apply Thm. 2.1 with the uniform prior $(\forall a, P(a) = \frac{1}{K})$ and $\lambda = \lambda_m$ (depending possibly on $m$): with probability $1 - \delta/2$, for any tuple $(m, \lambda_m)$ with $m \geq 1$, any posterior $Q$,

$$|M_m(Q)| \leq \frac{\mathrm{KL}(Q, P) + 2 + \log(4/\delta)}{\lambda_m} + \frac{\lambda_m}{2}\left(\hat{V}_m(Q) + V_m(Q)\right).$$

Notice that for any $Q$, $\mathrm{KL}(Q, P) \leq \log(K)$ by concavity of the log. We now fix an horizon $M > 0$, we then have in particular, with probability $1 - \delta/2$: for any posterior $Q$,

$$|M_m(Q)| \leq \frac{\log(K) + 2\log(k + 1) + \log(4/\delta)}{\lambda_k} + \frac{\lambda_m}{2}\left(\hat{V}_m(Q) + V_m(Q)\right).$$

We now have to deal with $V_k(Q), \hat{V}_k(Q)$ for all $k \leq m$. To do so, we propose the two following lemmas.

**Lemma C.1.** *For all $m \geq 1$, $a \in \mathcal{A}$, $V_m(a) \leq \frac{2Cm}{\varepsilon_m}$. Then, we have for any $m, Q$, $V_m(Q) \leq \frac{2Cm}{\varepsilon_m}$.*

*Proof.* We have

$$V_t(a) = \sum_{i=1}^m \mathbb{E}\left[\left(\left[R_i^{a^*} - R_i^a\right] - \Delta(a)\right)^2 \mid \mathcal{F}_{i-1}\right]$$

$$= \sum_{i=1}^m \mathbb{E}\left[\left(R_i^{a^*} - R_i^a\right)^2 \mid \mathcal{F}_{i-1}\right] - m\Delta(a)^2$$

$$\leq \sum_{i=1}^m \mathbb{E}\left[\left(R_i^{a^*} - R_i^a\right)^2 \mid \mathcal{F}_{i-1}\right]$$

$$= \sum_{i=1}^m \mathbb{E}\left[\mathbb{E}_{A_i \sim \pi_i} \mathbb{E}_{R_i}\left[\frac{1}{\pi_i(a^*)^2} R_i(a^*)^2 \mathbb{1}(A_i = a^*) + \frac{1}{\pi_i(a)^2} R_i(a)^2 \mathbb{1}(A_i = a)\right] \mid \mathcal{F}_{i-1}\right].$$

The last line holding because $R_i$ is independent of $\mathcal{F}_{i-1}$, $A_i$ is independent of $R_i$ and $\pi$ is $\mathcal{F}_{i-1}$ measurable. We now use that for all $i, a$, $\mathbb{E}_{R_i}[R_i(a)^2] \leq C$

$$
\begin{aligned}
&= \sum_{i=1}^{m} \mathbb{E}\left[\mathbb{E}_{A_i \sim \pi_i}\left[\frac{1}{\pi_i(a^*)^2}C\mathbb{1}(A_i = a^*) + \frac{1}{\pi_i(a)^2}C\mathbb{1}(A_i = a)\right] \mid \mathcal{F}_{i-1}\right] \\
&= \sum_{i=1}^{m} C\left(\frac{\pi_i(a)}{\pi_i(a)^2} + \frac{\pi_i(a^*)}{\pi_i(a^*)^2}\right) \\
&= \sum_{i=1}^{m} C\left(\frac{1}{\pi_i(a)} + \frac{1}{\pi_i(a^*)}\right) \\
&\leq \frac{2Cm}{\varepsilon_m}.
\end{aligned}
$$

$\square$

**Lemma C.2.** *Let $m \geq 1$, with probability $1 - \delta/2$, for any posterior $Q$, we have*

$$
\hat{V}_m(Q) \leq \frac{4CKm}{\varepsilon_m \delta}.
$$

*Proof.* Let $Q$ a distribution over $\mathcal{A}$. Recall that

$$
\begin{aligned}
\hat{V}_m(Q) &= \sum_{i=1}^{m}\left(R_i^{a^*} - R_i^a - [R(a^*) - R(a)]\right)^2 \\
&= \sum_{a \in \mathcal{A}} Q(a)\hat{V}_m(a).
\end{aligned}
$$

Notice that for any $a$, $(S\hat{M}_m^a)_m$ is a nonnegative random variable. We then apply Markov's inequality for any $a$, with probability $1 - \delta/2K$

$$
\hat{V}_m(a) \leq \frac{2K\mathbb{E}[\hat{V}_m(a)]}{\delta}.
$$

Noticing that $\mathbb{E}[\hat{V}_m(a)] = \mathbb{E}[V_m(a)]$, we can apply Lemma C.1 to conclude that

$$
\mathbb{E}[\hat{V}_m(a)] \leq \frac{2Cm}{\varepsilon_m}.
$$

Finally, taking an union bound on thoser events for all $a \in \mathcal{A}$ gives us, with probability $1 - \delta/2$, for any posterior $Q$

$$
\begin{aligned}
V_m(Q) &\leq \sum_{a \in \mathcal{A}} Q(a)\hat{V}_m(a) \\
&\leq \sum_{a \in \mathcal{A}} Q(a)\frac{4CKm}{\varepsilon_m \delta} \\
&= \frac{4CKm}{\varepsilon_m \delta}.
\end{aligned}
$$

This concludes the proof. $\square$

To conclude, we apply Lemmas C.1 and C.2 to get that with probability $1 - \delta$, for any posterior $Q$

$$
|M_m(Q)| \leq \frac{\mathrm{KL}(Q, P) + \log(4/\delta)}{\lambda_m} + \frac{Cm\lambda_m}{\varepsilon_m}\left(1 + \frac{2K}{\delta}\right).
$$

Dividing by $m$ and taking

$$\lambda_m = \sqrt{\frac{(\log(K) + \log(4/\delta))\, \varepsilon_m}{Cm\left(1 + \frac{2K}{\delta}\right)}}$$

concludes the proof.

$\square$

## C.3 Proof of Cor. B.1

*Proof.* Fix $\delta > 0$. For any pair $(\lambda_k, P_k), k \geq 1$, we apply Thm. 2.1 with

$$\delta_k := \frac{\delta}{k(k+1)} \geq \frac{\delta}{(k+1)^2}.$$

Notice that we have $\sum_{k=1}^{+\infty} \delta_k = \delta$. We then have with probability $1 - \delta_k$ over $S$, for any $m \geq 1$, any posterior $Q$,

$$|M_m(Q)| \leq \frac{KL(Q, P_k) + 2\log(k+1) + \log(2/\delta)}{\lambda_k} + \frac{\lambda_k}{2}\left(\hat{V}_m(Q) + V_m(Q)\right).$$

Taking an union bound on all those event, gives the final result, valid with probability $1 - \delta$ over the sample $S$, for any any tuple $(m, \lambda_k, P_k)$ with $m, k \geq 1$, any posterior $Q$ over $\mathcal{H}$. This gives Eq. (6).

To obtain Eq. (7), we restrict the range of Eq. (6) to the tuples $(m, \lambda_m, P_m), m \geq 1$ (the restricted set of tuples where $k = m$) and we bound both $\hat{V}_m(Q), V_m(Q)$ by $\sum_{i=1}^{m} C_i^2$ to conclude. $\square$

## C.4 Proof of Cor. B.2

*Proof.* For the first bound we start from the intermediary result Eq. (3) of Thm. 2.1. Using the same marrtingale as in Thm. 2.3 gives, for any $\eta \in \mathbb{R}$, holding with probability $1 - \delta$ for any $m > 0, Q \in \mathcal{M}_1^+(\mathcal{H})$

$$\eta\left(\sum_{i=1}^{m} \mathbb{E}_{h \sim Q}[\ell(h, z_i)] - m\mathbb{E}_{h \sim Q}[R(h)]\right) \leq KL(Q, P) + \log(1/\delta) + \frac{\eta^2}{2}\sum_{i=1}^{m} \mathbb{E}_{h \sim Q}[\Delta[M]_i(h) + \Delta\langle M\rangle_i(h)].$$

Taking $\eta = \pm\lambda$ with $\lambda > 0$ gives

$$\lambda m\left|\mathbb{E}_{h \sim Q}[R(h) - R_m(h)]\right| \leq KL(Q, P) + \log(1/\delta) + \frac{\lambda^2}{2}\sum_{i=1}^{m} \mathbb{E}_{h \sim Q}[\Delta[M]_i(h) + \Delta\langle M\rangle_i(h)]. \qquad (8)$$

Finally, divide by $\lambda m$ and bound $\Delta[M]_i(h) + \Delta\langle M\rangle_i(h)$ by $2K^2$ to conclude.

For the second bound, we start from Eq. (8) again and for a fixed $m$, we now apply our result with $\lambda' = \lambda/m$. We then have for any $m$, with probability $1 - \delta$, for any $Q$

$$\lambda\left|\mathbb{E}_{h \sim Q}[R(h) - R_m(h)]\right| \leq KL(Q, P) + \log(1/\delta) + \frac{\lambda^2}{2m^2}\sum_{i=1}^{m} \mathbb{E}_{h \sim Q}[\Delta[M]_i(h) + \Delta\langle M\rangle_i(h)].$$

Finally, dividing by $\lambda$, bounding $\Delta[M]_i(h) + \Delta\langle M\rangle_i(h)$ by $2K^2$ and rearranging the terms concludes the proof. $\square$

