# OpenReview forum: "PAC-Bayes Generalisation Bounds for Heavy-Tailed Losses through Supermartingales"
_TMLR — Accepted by TMLR_

### Review · Reviewer_Foq3 · 2023-02-24

**Summary Of Contributions:**

Broadly speaking, this is a technical paper which looks at extending the PAC-Bayes analytical machinery to new learning problems and algorithms. More precisely, they are interested in problems involving (random) losses or rewards that are allowed to be unbounded, and potentially dependent (up to martingale requirements).

My understanding of the main contributions is as follows.

As a foundational technical lemma, they exploit Ville's maximal inequality (extending beyond the Markov inequality step typical to PAC-Bayes proofs) to deal with martingale difference sequences that are potentially unbounded. The main general-purpose result here is Thm. 2.1. Martingales in the PAC-Bayes setting were addressed in previous work by Seldin et al. a decade ago, but with boundedness assumptions. The "corollary" of this result is Thm. 2.3, which gives a more typical PAC-Bayes learning setup (iid data, expected loss bounds), and does not require bounded losses. Their bound here involves quadratic loss terms (both sample and population) and holds jointly over all sample sizes. The dependence on $\delta$ is better than that of Alquier and Guedj (2018), and they do not have terms that can increase with sample size as in Holland (2019). As for Kuzborskij and Szepesvari (2019; henceforth KS19), the result is very similar, albeit looser (since $\lambda$ cannot be optimized in the same way here), but holds for any prior.

In addition, they build upon the recent Online PAC-Bayes framework of Haddouche and Guedj (2022), essentially leaving the framework as-is, but removing the need for sub-Gaussianity (instead using quadratic deviations), and obtaining bounds that hold over all time steps.



**Audience:**

Yes

**Broader Impact Concerns:**

No broader impact concerns.

**Claims And Evidence:**

Yes

**Requested Changes:**

Below, I will highlight a few specific points that I tripped up on while reading this paper; it is not a comprehensive list, but I think the authors can generalize beyond these points easily, and that doing so would strengthen the paper.

*Section 2.1 is unclear*
- The overall flow of this section is rather unnatural. The main result Thm. 2.1 is buried between "Framework" and "Analysis of the bound" paragraphs; the proof also starts abruptly after the literature comparison. Why not break the proof into a separate sub-section? (I think the same should be done for the main result of section 3 as well)
- I found the "Comparison with literature" quite terse. This is an important paragraph, comparing the work with that of Seldin et al., but the price paid to be able to handle unbounded random variables is not clear. The authors mention "Bounded variations of the martingale difference sequence", "restrictions on the range of $\lambda$" and "Bernstein", but I think many readers will not know what the authors are talking about - the only assumption in this section is that of a martingale difference sequence and a data-free prior. Presumably a reference to certain points in section 1.2 or the proof of Thm 2.1 could clear things up.

*Claims in 2.2:*
The comparison with KS19 is mentioned briefly in the last paragraph with details relegated to the appendix, but I think this is worth the effort of explaining to the reader, since KS19 gives PAC-Bayes bounds for unbounded losses. Related to this, on p.7, the authors say that their bound "matches the usual PAC-Bayes literature" (which is true), "while for unbounded losses, which is novel" (this is questionable). I assume the authors have the "holds jointly over all $m$" qualifier in mind when they make this statement, but it is confusing and I think this paragraph should be re-written. The proof route here is very different from KS19, and it is rooted in a result that allows for dependent random variables.

*Notation:*
When introducing the *stochastic kernel* on p.8, they introduce $\mathcal{S}$; does this appear anywhere else? Note also the mapping $Q$ (same paragraph) maps to "$[0;1]$", I think the unit interval is usually written $[0,1]$.

*Bandit extension:*
In section 4, the authors essentially match the Seldin et al. bounds for multi-armed bandits without requiring bounded rewards, though with the cost being "localization". This is the opposite of what we see in the batch iid case, where the authors' bounds hold simultaneously over all sample sizes; unless I have missed it, I think a paragraph describing the reason for this difference would be beneficial to the reader.

**Strengths And Weaknesses:**

*Strengths:*
- The basic problem of interest is stated clearly, with comprehensive references from the literature giving their results context.
- The technical strategy taken in this paper is explained quite clearly, as is the impact this strategy has on the resulting PAC-Bayes style error bounds.

*Weaknesses:*
Overall, I found the presentation and writing to be somewhat sloppy. I think there is a fair bit of room for improvement here.

---

> ### Author Response · Authors · 2023-03-16
> **Thank you for your review, answers to your concerns.**
>
> We thank you for your careful review and are glad to read that our basic problem and technical strategy are 'stated clearly'. We agree that our presentation would have gained to be clearer in Section 2,3 and 4. We state below our main changes, dedicated to making our contributions and comparisons clearer.
>
> - Section 2.1: Thank you for your comments, we highlighted our main theorem by adding (in red) a new paragraph and a few lines to introduce the result. We also detailed more carefully the comparison with Seldin et al. 2012. We precisely stated the assumption made in their work and explained why we were able to avoid it and at what cost (i.e. an additional empirical moment). We also put the proof of our main result in a separated subsection for more clarity.
> - Section 2.2: Similarly, we highlighted the main result of this section. We precise that there is no supplementary comparison with K&S 2019 in the appendix: the whole comparison is stated in Section 2.2. We have detailed more carefully our comparison to make the reader understand what precisely are our contributions with respect to K&S 2019. We have also rewritten some part of the paragraph 'About the choice of $\lambda$' for the sake of clarity.
> - Section 3: We split the section in two parts: the first one including the main result as well as discussions, the latter detailing the proof. We also provided several small modifications for pedagogical purposes, all detailed in red.
> - Section 4: we detailed more precisely why in its current form, Thm 4.1 is not suited to take into account all times simultaneously.
>
> We hope all those adjustments have made our work more pedagogical and readable.

---

### Review · Reviewer_bQp1 · 2023-03-06

**Summary Of Contributions:**

This paper provides a new concentration inequality for parametrized martingales $M_1(h), M_2(h),\dots$ where $h\in H$ is some parameter (often $H$ is a hypothesis class but in principle it could be anything). Specifically, for any distributions $P$ and $Q$ over $H$, we define $M_n(Q)=\mathbb{E}_{h\sim Q}[M_n(h)]$ and then it is shown that for any $\lambda, \delta$, with probability $1-\delta$, for all $n$:

$|M_n(Q)| \le \frac{KL(Q,P) + \log(2/\delta)}{\lambda} + \frac{\lambda}{2} V_n$

Where $V_n$ is some technical analog of variance that we might expect to be $O(n)$ in the case that $M_n$ is bounded or has light tails, but might be controllable in other cases as well. Notably, by taking a union bound over pairs $(\lambda, \delta) = (2^i, 6\delta /\pi i^2)$ for all positive integers $i$, one recovers an analog of the law of iterated logarithm (this fact seems not mentioned in the paper, but I think it might be worth a remark).

Importantly, the result does  not require any light tailed conditions on the martingales and so can be used to generate concentration inequalities that require only a bound on the second moment. This appears to be “paid for” in the term $V_n$, which now depends not only on the second moment of $M$, but also on an “empirical” second moment.

This main result is used to extend several previous results about generalization error, online learning, and bandits to the case of unbounded losses.



**Audience:**

Yes

**Claims And Evidence:**

Yes

**Requested Changes:**

Please address the issues raised in the review. Adding more discussion as suggested in various places would help the paper IMO, but is not required for securing my recommendation.

**Strengths And Weaknesses:**

Strengths:
The result is new to my knowledge, and the proof technique is very clean: up to the technical Lemma 2.2 which is essentially a careful verification that one can interchange certain integrals, the proof follows the method of building a submartingale $Z_n(\eta)$ from a martingale $M_n$ by setting $Z_n(\eta) = \exp(\eta M_n - \eta^2 V_n)$ and then using Ville’s maximal inequality to bound $Z_n(\eta)$ for all $n$.

The applications to previous results are interesting, although to my mind the basic main theorem is the most valuable part of the paper. Indeed, the application to multi-armed bandits seems to actually have the wrong dependence on the number of arms $K$ Although the authors acknowledge that the bandit result is probably not tight, one has to inspect the bound carefully to see this worse dependence - I think the paper would be better to actually mention this to guide future readers to future questions.

Weakness:

This is not much of a weakness but more a comment:
Whether having the dependence on the empirical variance in Theorem 2.1 is wholly a negative is up for debate: the empirical second moment is observable and so is arguable more useful than the true second moment.The authors attempt to exploit this in equation (4) which provides a kind of analog of a “structural risk minimization” objective using empirically observable quantities. However, it seems unsatisfying for  two reasons:
1. The dependence on the actual variance is ignored, so as far as I can tell, we cannot conclude what  the relationship is between solutions to this objective and the generalization error.
2. The value of $\lambda$ is fixed. The authors seem to claim this is a feature rather than a bug, but I am not convinced. Surely it would be better to have a data-dependence $\lambda$ (intuitively it should be something like $1/\sqrt{\text{some variance term}}$. I do not understand what is meant by a fixed $\lambda$ being helpful  ”in order to ensure a stable  learning algorithm”. Stable in what sense? The only argument I can see for having a non-adaptive $\lambda$ is computational as the objective might be harder to solve. In particular with a tuned $\lambda$, the result should presumably become a kind of "generalization bound analog" of adaptive algorithms in the learning with expert advice literature in which the  ideal algorithm obtains regret $\tilde O(\sqrt{KL(P,Q)  V_n})$ where $P$ is  some prior, $Q$ is a comparator and $V_n$ is  variance-analog just as in this paper  (e.g. see https://arxiv.org/pdf/2102.01046.pdf and references therein).

These seem like they should be fixable if it were possible to obtain a purely empirical bound with only $[M]_(Q) $ without $\langle M\rangle_m(Q)$. After tuning $\lambda$, the objective might become more difficult to minimize of course, but that seems a secondary issue.

Minor comments:
Lemma 1.3: I think you have not defined $[M]_m$ and $\langle M\rangle_m$ yet - just $\Delta[M]_m$ and $\Delta\langle M\rangle_m$. Please add the definitions here.

The authors might take a look at recent papers https://arxiv.org/abs/2110.14099 and https://arxiv.org/abs/2302.05829, which use similar techniques to prove bounds on martingales. The latter in particular appears to be a similar result (although contemporaneous so I do not have novelty concerns on that front).

---

> ### Author Response · Authors · 2023-03-16
> **Thank you for your review, we added discussions and clarifications in the revised version**
>
> We thank you for your review and are happy to read that you found our proof technique 'clean'. We added several new discussions (in red in the revised version) throughout our work to make it more pedagogical. We address below your concerns.
>
> - We elaborated (in red) on the bandits bound in Section 4, to make appear its weaknesses (and the reason behind those) more explicitly.
> 1. It is true that the dependency on the theoretical variance is not taken into our optimisation objective, the reason being that such a variance is not accessible to the learner in practice. Thus, our learning objective implies that we are able to bound the true variance by a known constant: this is what we meant in the abstract when we said that 'PAC-Bayes provides generalisation bounds for unbounded losses with the sole assumption of bounded variance of the loss function'. We pressed this point in red in the main document.
> 	This point is not new in PAC-Bayes: for instance, the classical PAC-Bayes learning objective, derived from Theorem A.2 (holding for a loss bounded by $K$) is the following :
> 	$\operatorname{argmin_{Q}}  \mathbb{E}_{h\sim Q} \left[\frac\{1\}\{m\}\sum_\{i=1\}^m\ell(h,z_i)\right] + \frac\{\operatorname\{KL\}(Q,P)\}\{\lambda\}$  and does not take into account the term $\lambda K^2/2m$ .
> 2.  We agree that considering an hyperparameter $\lambda$ is not essential theoretically speaking. Indeed, we acknowledge for instance, when comparing to Kuzborskij & Szepesvári 2019 that their result is tighter than ours (in the case of independent data only) as their bound is similar to ours when being optimised in $\lambda$. The term 'stable' was also ill-chosen and we reformulated in red our point, which stands mainly from a computational point of view: in practice, using a fixed hyperparameter $\lambda$ (acting as an analoguous of a learning step) makes the learning algorithm easier to run and this technique is often used in practice (usually with a fine-tuning of $\lambda$ over a discrete grid).
> 	We also were not aware of the reference you mentioned and the possible links to the experts advice literature, it seems to be a promising lead for future work but could be hard to attain as in PAC-Bayes the number of 'experts' is often uncountable.
>
> We added the required definition in Lemma 1.3, thanks for pointing this out.
> Finally we thank you for the additional references, we were not aware of those works earlier and we now cite the coin-betting PAC-Bayes work in our conclusion.
>
> We hope this addresses your concerns.

---

### Review · Reviewer_oC3Q · 2023-03-07

**Summary Of Contributions:**

This paper studies the problem of PAC-Bayes bounds under the assumption that the losses can be unbounded. The well-known results for PAC-Bayes mainly stated for bounded loss functions.

The authors in this paper provide a nice technique for obtaining Pac-Bayes bound for unbounded losses. They also show the applications of their technique for statistical learning with iid data and PAC-bayes guarantee for online leanring.

**Audience:**

Yes

**Broader Impact Concerns:**

There is no concern in this front.

**Claims And Evidence:**

Yes

**Requested Changes:**

I think the paper is good as is.

**Strengths And Weaknesses:**

I found the paper very well-written and the stated lemma in the main body helps the reader understand the main contribution of the paper. I think the technique used in the paper is novel and can have a lasting impact on the literature.

---

> ### Author Response · Authors · 2023-03-16
> **Many thanks for the positive feedback**
>
> We warmly thank you for your positive review, we are glad to read that you found our paper 'very well written' and that you believe our work could be impactful.
>
> We thank you for your time.

---

### Decision · Action_Editors · 2023-04-09

**Recommendation:** Accept with minor revision

**Comment:**

Three reviewers read the manuscript. Even in the initial reviews, the reviewers were unanimous that the technique is new and the results are important. I agree with them.

On the other hand, the reviewers disagreed on the writing of the initial version. While oC3Q "found the paper very well-written", bQp1 found it "acceptable" and provided a list of minor problems to be fixed. Foq3 wrote "I found the presentation and writing to be somewhat sloppy. I think there is a fair bit of room for improvement here" and pointed out many parts of the paper that were unclear or inaccurate. I thank reviewers bQp1 and Foq3 for their detailed reading of the paper and their constructive comments. These were taken into accounts by the authors and it was very beneficial to the paper. About the new version, Foq3 wrote "I feel that the paper presents reasonable claims with evidence presented in a sufficiently clear form, and that the paper should have an audience if included in TMLR" and bQp1 wrote "the authors have improved the readability in the revision, and I am happy to recommend acceptance".

Thus, I am also happy to support the publication of the paper in TMLR, and I sincerely congratulate the authors for this nice work.

That being said, I maintain that both the abstract and the introduction still contain inaccuracies on the subexponential/bounded problem. This is both unfair to some existing works, and misleading for future readers: sub-exponentiality is not boundedness. Thus, I will require a minor revision in order to fix these issues.

More precisely:

1) PAC-Bayes bounds for unbounded losses are quite old. In Catoni's St-Flour notes (2021), Chapter 5 is dedicated to PAC-Bayes bounds for classification AND regression with quadratic loss, under the assumptions that the loss is sub-exponential (Equation 5.8.1 page 181). While this is still restrictive, it allows to deal with Gaussian noise, which is I believe an important example of unbounded noise. This approach was reused in a large number of papers on PAC-Bayes since. For example, Guedj's PhD thesis [Aggregation of estimators and classifiers : theory and methods, 2013] studies nonparametric additive regression under the assumption that the noise is subexponential (Assumption 2.1 page 40) and indeed follows Catoni's technique in the proofs. For these reasons, I think the following sentence is misleading: "Several works (Alquier and Guedj, 2018; Holland, 2019; Haddouche et al., 2021) already proposed routes to overcome the boundedness constraint". There are two problems in one sentence:
- PAC-Bayes bounds for unbounded losses are standard in the sub-exponential case, and Catoni's lecture notes should be cited,
- the main contribution of (Alquier and Guedj, 2018; Holland, 2019) was not to tackle the unbounded case, but heavy-tailed losses (or without exponential moments). Alquier and Guedj (page 889): "This paper shows that a proof scheme of PAC-Bayesian bounds proposed by Bégin et al. (2016) can be extended to a very general setting, without independence nor exponential moments assumptions". Holland (abstract): "We derive PAC-Bayesian learning guarantees for heavy-tailed losses".

I will ask the authors to fix these sentences. I would even suggest to fix the title: "PAC-Bayes bounds for Heavy-tailed Losses through Supermartingales" instead of "PAC-Bayes bounds for Unbounded Losses through Supermartingales", as the current title is not an accurate description the contribution of the paper, but I will leave the choice to the authors.

2) there were other works on PAC-Bayes bounds with heavy-tailed losses besides (Alquier and Guedj, 2018; Holland, 2019). The paper [Audibert and Catoni, Robust linear least squares regression, AOS, 2011] proves PAC bounds for least square estimators with unbounded (and potentially heavy-tailed) X's and noise. While the estimator studied is not Bayesian, the proof relies on PAC-Bayes bounds and the results could completely be extended to Bayesian estimators. A problem in the proof makes the result suboptimal with respect to the dimension, this was later fixed by [Catoni and Giulini, Dimension-free PAC-Bayesian bounds for matrices, vectors, and linear least squares regression, arXiv, 2017] and [Catoni, PAC-Bayesian bounds for the Gram matrix and least squares regression with a random design, arXiv, 2018]. The latter proves PAC-Bayes bounds for the quadratic loss under the only assumption that moments of order 4 are finite.

Thus, the comment in the abstract "(as simple as the squared loss on an unbounded space)" is particularly poorly chosen: results in this case have been known for years. It is very important to remove this comment from the paper before publication and I think it would be honest to cite Catoni and Giulini's papers. As Catoni and Giulini's technique is very specific to the quadratic loss, this does not remove any interest from the new supermartingale technique.

**Audience:**

The paper is more theoretical than the average TMLR paper. The contributions will probably be more appealing to theoreticians. However, given their recent success to provide tight generalization certificates for deep networks, PAC-Bayes bounds should be of intrest to many TMLR readers.

**Claims And Evidence:**

PAC-Bayes bounds are PAC bounds on aggregated or randomized algorithms. The original papers focused on classifications and the bounds proven hold only for bounded loss functions. The authors state in the abstract that the "extension
to the case of unbounded losses (as simple as the squared loss on an unbounded space) remains largely uncharted". This statement is an exaggeration, especially as the authors do not mention classical PAC-Bayes bounds under sub-exponential losses in their paper (see below). The difficulty is not to extend PAC-Bayes bounds beyond the bounded case, but rather beyond the sub-exponential case (aka heavy-tailed case). That being said, the authors do a pretty good job in the heavy-tailed case. The authors propose a new way to derive PAC-Bayes bounds based on Ville's maximal inequality for supermartingales. As a consequence, in the batch setting, their result improves on the existing PAC-Bayes bounds for heavy-tailed losses. In the online setting, they extend their own NeurIPS 2022 paper that required exponential moments.

---

> ### Author Response · Authors · 2023-04-20
> **Thank you for your decision, we answered your concerns in the newest version.**
>
> We thank you for your decision and for the time you took spot our approximations.  Thank you to give us the final decision of the reviewers as we are not able to see their latest answer.
>
> I was personally not aware of the new references you mentioned and I thank you for those corrections as the introduction is designed to be as pedagogical and comprehensive as possible and the distinction between unbounded losses and light-tailed ones in PAC-Bayes, while often eluded in the modern literature, was an important point we missed.
>
> We hope the corrected version of the document (changes stated in red in the introduction) answered you points of inquiry and we are happy to modify again our manuscript if needed.
>
> We thank you again for your time and for conducting the reviewing process.